# Cold Sintering Isomagnetic Dielectric NaCl-Nickel Zinc Ferrite Composite Ceramics

Jiuyuan Han [1], Mengjiao Chi [1], Liu Yang [1,*], Bing Liu [1], Minmin Mao [1], Hadi Barzegar Bafrooei [1], Zhongyan Ma [2], Yingjie Ren [2], Feng Shi [3], Ehsan Taheri-Nassaj [4], Dawei Wang [5,*] and Kaixin Song [1,*]

[1] College of Electronics Information, Hangzhou Dianzi University, Hangzhou 310018, China; hanjiuyuanabc@163.com (J.H.); cmjzgzqcjls555@sina.com (M.C.); liubing@hdu.edu.cn (B.L.); mmm@hdu.edu.cn (M.M.); hadi@hdu.edu.cn (H.B.B.)

[2] Institute of Communication Materials, Zhejiang Wazam New Materials Co., Ltd., Hangzhou 311121, China; zhongyan.ma@hzccl.com (Z.M.); ryj1981@126.com (Y.R.)

[3] School of Materials Science and Engineering, Qilu University of Technology (Shandong Academy of Sciences), Jinan 250353, China; sf751106@sia.com.cn

[4] Department of Materials Engineering, Faculty of Engineering, Tarbiat Modares University, Tehran 14115-143, Iran; taheri@modares.ac.ir

[5] Functional Materials and Acousto-Optic Instruments Institute, School of Instrumentation Science and Engineering, Harbin Institute of Technology, Harbin 150080, China

[*] Correspondence: 40522@hdu.edu.cn (L.Y.); wangdawei102@gmail.com (D.W.); kxsong@hdu.edu.cn (K.S.)

**Abstract:** In this study, dense composites of $x$NaCl-$(1-x)$Ni$_{0.5}$Zn$_{0.5}$Fe$_2$O$_4$ (referred to as NaCl-NZO) and $x$H$_3$BO$_3$-$(0.8-x)$Ni$_{0.5}$Zn$_{0.5}$Fe$_2$O$_4$-0.2NaCl (referred to as HB-NZO-NaCl) were prepared using the cold sintering process. The objective was to investigate the cold sintering procedure for fabricating ferrite composite ceramics with comparable permeability and dielectric constants suitable for radio-frequency electronic device applications. Optimal cold sintering conditions were determined as 200 °C/30 min/500 MPa with a relative density of 95% for NaCl-NZO and 120 °C/30 min/300 MPa with a relative density of 95.4% for HB-NZO-NaCl. X-ray diffraction and scanning electron microscope analyses confirmed the absence of secondary phases. The resulting composite ceramics showed promising characteristics, with the 0.5NaCl-0.5NZO composition exhibiting a dielectric constant of 6.2 @ 100 MHz, dielectric loss of 0.02 @ 100 MHz, permeability of 2.5 @ 100 MHz, and magnetic loss of 0.001 @ 100 MHz. Similarly, the 0.3HB-0.5NZO-0.2NaCl composition displayed a dielectric constant of 5.9 @ 100 MHz, dielectric loss of 0.02 @ 100 MHz, permeability of 5.1 @ 100 MHz, and magnetic loss of $5 \times 10^{-4}$ @ 100 MHz. These findings indicate potential applications in wireless communication.

**Keywords:** cold sintering; magnetic–dielectric materials; ferrite; dielectric constant

## 1. Introduction

In recent years, due to the rapid development of science and technology, electronic products have been continuously developing towards miniaturization, intelligence, and multifunctional directions. As such, the integration of products is also required to be increasingly high, which poses a huge demand for miniaturization of devices. The Very High Frequency(VHF) band is widely used for disseminating traffic management information systems (VTS) and providing visual range communication for sound and data for civil aircraft. Due to the frequency characteristics of VHF, research on the miniaturization of antennas belonging to transmitting and receiving channels is constantly deepening. Isomagnetic dielectric materials play a crucial role in antenna miniaturization due to their electromagnetic properties especially for VHF antenna modules. To achieve a high magnetic–electric coupling coefficient, researchers have developed magneto–electric composite materials by combining ferroelectric and ferrite materials [1,2]. However, these composite materials often suffer from drawbacks, such as high dielectric and magnetic losses, as well as poor frequency stability, which limit their applicability in the high-frequency range. To address

these issues, a two-phase approach involving magnetic and dielectric materials has been employed to develop new isomagnetic–dielectric composites with high dielectric constants, strong magnetic conductivity, and low losses [3–7].

Isomagnetic dielectric material is a type of material which has both magnetic permeability and dielectric constant, and the value of them are almost equal. It is beneficial for the discrete devices because of its high resistivity, which can reduce eddy current loss. When used as an antenna substrate, the characteristic impedance of the antenna substrate material is equal to the characteristic impedance of the vacuum, and the characteristic impedance of the vacuum is basically equal to the characteristic impedance of the air, which means that the antenna will reflect very little when radiating energy into free space, which is also beneficial to improving the radiation efficiency of the antenna. Additionally, it is expected that the material will have a large equimagnetic dielectric bandwidth for widespread application. Previous studies have attempted to prepare composite ceramics by co-sintering high dielectric constant ceramics with various ferrites [8–10]. However, these efforts have resulted in significant disparities in the dielectric constants and magnetic permeability of the composite ceramics, making it challenging to match the device's impedance with that of free space. Furthermore, high-temperature sintering of these ceramics has led to phase transformations and interface effects between the dielectric ceramic and the ferrite, reducing the predictability of the final samples. Additionally, the mismatched thermal expansion between different materials has resulted in reduced durability and complex preparation processes. Consequently, traditional sintering techniques have struggled to produce structurally dense composite materials with stable performance [11,12]. Therefore, there is a critical need to develop a new method for low-temperature (<300 °C) composites of dielectric ceramics and ferrite materials, where the ferrite particles can be uniformly dispersed within the high-density dielectric ceramics, resulting in desirable magnetic–dielectric properties.

NaCl, with its excellent dielectric properties ($\varepsilon$r = 5.55 and Q × f = 49,600 GHz) and a density of 2.165 g/cm$^3$, has been identified as a promising candidate for dielectric ceramics. Dry-pressed NaCl ceramics can achieve relative densities of 94.3–94.6%. Cold sintering technology has successfully been utilized to achieve densification of ceramics, such as $Al_2O_3$-NaCl, $SrFe_{12}O_{19}$-NaCl, and $MgTiO_3$-NaCl [13–16], demonstrating the feasibility of densifying NaCl ceramics using cold sintering. Similarly, $H_3BO_3$ (HB) ceramic has been successfully densified at room temperature, exhibiting a dielectric constant ($\varepsilon$r) of 2.83 and a relative density of 97.6% [17]. Notably, $H_3BO_3$ has been recognized for its low-loss characteristics, with a dielectric constant of 2.84 and a bulk density of 1.46 g/cm$^3$ [18].

In this study, we successfully fabricated two dense composite materials of xNaCl-(1−x)$Ni_{0.5}Zn_{0.5}Fe_2O_4$, which is abbreviated as NaCl-NZO during the following discussion, and x$H_3BO_3$-(0.8−x)$Ni_{0.5}Zn_{0.5}Fe_2O_4$-0.2NaCl (abbreviated as HB-NZO-NaCl) using cold sintering technology. By overcoming the limitations of sintering temperature, this approach expands the potential market for this series of ceramics, aligning with the goals of energy efficiency and emission reduction.

## 2. Experimental Procedure

Initially, all raw materials, including NiO (99.99%, Aladdin), ZnO (99.99%, Aladdin), $Fe_2O_3$ (99.99%, Aladdin), NaCl (99.99%, Aladdin), and $H_3BO_3$ (99.99%, Aladdin), were subjected to a 24 h drying process in a drying oven to eliminate atmospheric moisture. Following the chemical stoichiometric ratio of the $Ni_{0.5}Zn_{0.5}Fe_2O_4$ formula, ball milling was performed for 12 h using anhydrous ethanol as the medium to achieve uniform mixing and to refine the particle size of the raw powder. Subsequently, the resulting mixing slurry was dried at 80 °C. The powder was then calcined at 800 °C for 4 h, cooled to room temperature, and subjected to a second round of ball milling to further refine the particle size range of the pre-burned powder. This process aimed to enhance the reaction rate and to reduce the reaction time and temperature [19]. The slurry obtained from the ball milling underwent drying, and the powder was synthesized through solid-state sintering at

1080 °C. In this paper, two different composition types, namely $xNaCl$-$(1-x)Ni_{0.5}Zn_{0.5}Fe_2O_4$ and $xH_3BO_3$-$(0.8-x)Ni_{0.5}Zn_{0.5}Fe_2O_4$-$0.2NaCl$, were then mixed in different ratios using 8 wt% deionized water as the solvent. The composite powder and solvent were thoroughly ground in a grinding pot, passed through a 100-mesh sieve, and the uniformly mixed powder was poured into a steel mold for the cold sintering process.

As for cylindrical sample, weighed 1.5000 g ($\pm$0.0005 g) of powder and poured it into a 12.7 mm diameter mold before placing this into a hot press. Firstly, applied a certain uniaxial pressure (500 MPa) at room temperature for 10 min to promote particle rearrangement under pressure and liquid phase action. Then, increased the temperature to the specified temperature at a heating rate of 4 °C/min, maintained it at 200 °C for 30 min, and then lowered it to room temperature at the same rate. Finally removed the mold. Similarly, for the preparation of circular samples, weighing 1.000 g ($\pm$0.0005 g), with an outer diameter of 12.7 mm, an inner diameter of 7 mm, and a thickness set at 3 mm. The process was the same as above. Eventually removed the sample from the mold and let it dry in a constant temperature drying oven at 65 °C for 24 h to remove any residual liquid phase from the sample. All samples were prepared under the same conditions.

The phase composition of the samples was determined using an X-ray diffractometer (XRD, DX-2700, Haoyuan Co., Qingdao, China) with Cu K$\alpha$ radiation. The microstructures of the samples were characterized using a scanning electron microscope (SEM, ZEISS, Sigma 300, Aachen, Germany). The complex magnetic permeability and dielectric permittivity were measured using an Agilent 4991 impedance analyzer (Agilent Technologies, Palo Alto, CA, USA). The direct current (DC) resistivity was tested using precision power supply (Agilent, B2912A) with a resolution of 10 fA/100 nV from $-50$ V to $+50$ V.

## 3. Results and Analysis

Figure 1a illustrates the bulk and relative density curves of the composite ceramics prepared under different uniaxial pressures for 10 min. Then, all samples were kept at a sintering temperature of 200 °C for 30 min. The application of appropriate uniaxial pressure during the cold pressing stage before sintering enabled compaction of the powder particles and a reduction in voids and pores. At a uniaxial pressure of 500 MPa, the density of the $xNaCl/(1-x)NZO$ composite ceramics reached 2.86 g/cm$^3$, corresponding to a relative density of 94.8%. This result suggests that adding pressure during the sintering process enhances the mass transfer of liquid phase and increases the contact area between liquid phase and particles, thereby improving compactness. As the temperature rises, the liquid phase uniformly wets the particle surface, facilitating crystal growth due to capillary forces. Consequently, density will be improved due to the reduction in porosity [20]. However, surface cracking and decreasing density were found when the pressure was further increased to 600 MPa, which indicates that excessive pressure will disrupt the ceramic structure by reducing the contact area between the liquid phase and ceramic particles, thereby inhibiting sample compactness.

Figure 1b presents the bulk and relative density curves of the composite ceramics prepared under different cold sintering temperatures (30 min). Here, 10 min 500 MPa uniaxial pressure was selected based on the results in Figure 1a. It is evident that relative density significantly increases as the temperature rises from 120 °C to 140 °C. This can be attributed to the acceleration of chemical reactions during the sintering process and more compact arrangement of crystal particles facilitated by increased temperature [21]. The relative density continuously rises as the temperature further increases, demonstrating that higher temperatures accelerate the thermal activation mechanism in the sintering process. This, in turn, promotes chemical reactions by absorbing energy, exciting molecular and ionic movement, increasing particle surface activity, and strengthening the bonds between particles to enhance compactness [22]. Figure 1c investigates the influence of NaCl content on compactness. This group was tested under 500 MPa uniaxial pressure for 10 min, and then the sintering temperature was chosen to be 200 °C, according to Figure 1b. It can be observed that proper increase in proportion of low-density NaCl led to a density

decrease. However, with a further increase in NaCl content, relative density shows a slight increase followed by a decrease at the content of 0.5 wt%. This suggests that the NaCl content no longer improves compactness beyond this point. Therefore, the optimal cold sintering conditions are set at 200 °C/30 min/500 MPa. Finally, Figure 2 displays the XRD patterns of the xNaCl/(1−x)NZO composite materials, revealing the presence of NaCl and $Ni_{0.5}Zn_{0.5}Fe_2O_4$ phases without a third phase.

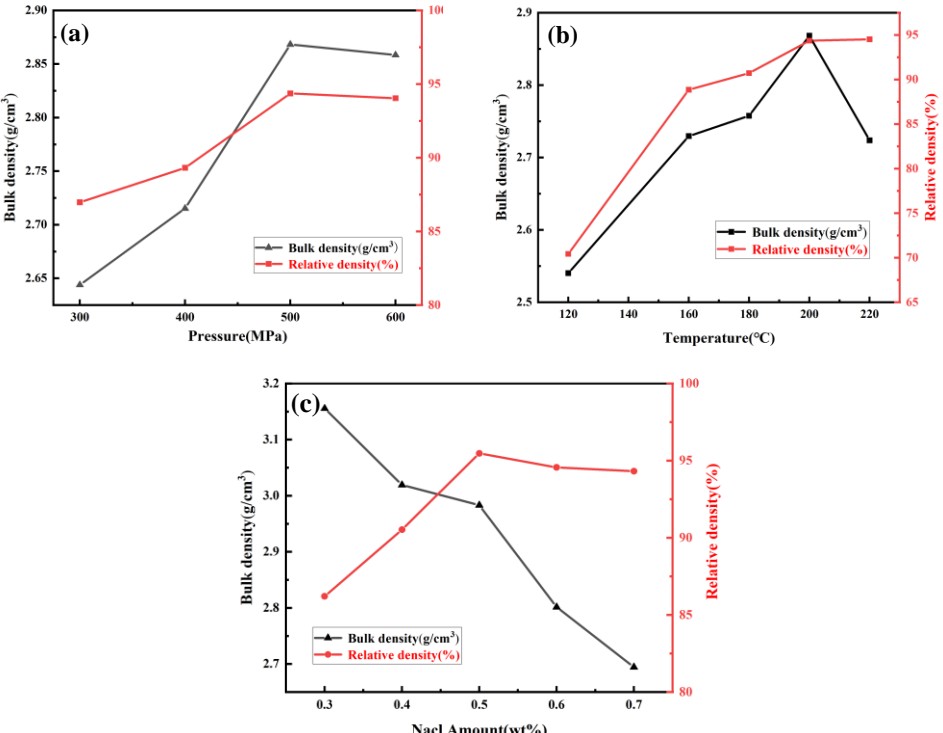

**Figure 1.** The density and relative density of xNaCl-(1−x)NZO composite ceramics as a function of different cold sintering conditions: (**a**) uniaxial pressure; (**b**) sintering temperature; (**c**) NaCl content.

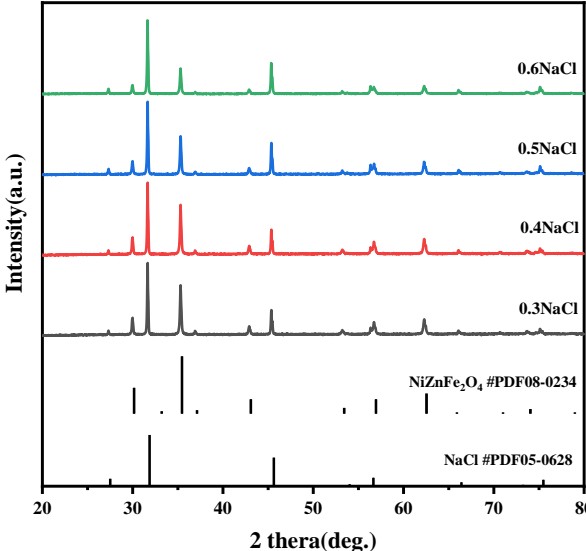

**Figure 2.** XRD pattern of multiphase ceramics with different NaCl contents under sintering conditions of 200 °C/30 min.

Figure S1 shows the SEM images of the sectional surface of xNaCl/(1−x)NZO samples with x = 0.3, 0.4, 0.5, and 0.6, respectively. Under the action of hot pressure, NaCl dissolved

in water will diffuse into the gap between NaCl particles and NZO particles to improve its density. Figure S2 shows the elemental distribution images of 0.5NaCl/0.5NZO composite ceramics prepared by cold sintering at 200 °C/30 min/500 MPa. Insoluble NZO particles are distributed around dissolved NaCl, which is because under external conditions, the "solution-precipitation" mechanism promotes mass transport and densification [14,15,17]. If excessive solvent is added, there will be pores in the sample after drying, which will also affect relative density.

The X-ray diffraction pattern and scanning electron microscope results of HB-NZO-NaCl samples are presented in the supplementary data. Figure S3 shows that the diffraction peaks of the NaCl phase, $Ni_{0.5}Zn_{0.5}Fe_2O_4$ phase, and $H_3BO_3$ phase all exist. The peak of the NaCl phase remains relatively stable during the process, consistent with the standard card peak (PDF05-0628), and there is no $HBO_2$ phase usually generated by the decomposition of $H_3BO_3$ [16]. This result shows that the phase compositions of powders are not changed during the cold sintering process. At the same time, it can be seen from the SEM (Figure S4) that all samples are dense, and the grain size and distribution are different from the previous group. These differences will have a certain impact on the microwave dielectric properties of dielectric ceramics and will be mentioned in the following discussion.

Figure 3a depicts the frequency-dependent dielectric constant of xNaCl-(1−x)NZO ceramics with varying NaCl contents. It is evident that the dielectric constant decreases with increasing x value in the low-frequency range from 10 MHz to 1000 MHz. This behavior is usually called dielectric dispersion, and results from the influence of dielectric polarization mechanisms on the dielectric constant at low frequencies [23]. Dielectric dispersion is typically attributed to the introduction of a non-centrally symmetric structure caused by the deviation of metal ions or molecules from their equilibrium position under an external electric field, leading to the spontaneous polarization of electric dipoles [24]. Furthermore, the dispersion behavior is also attributed to the presence of NZO grains with higher conductivity dispersed in the NaCl material with higher resistance. A higher NaCl content results in increased resistance, causing charge localization under the influence of the electric field and leading to interfacial polarization. However, this interfacial polarization does not follow the frequency of the applied electric field, resulting in a decrease in the dielectric constant. Figure 3b illustrates the unstable dielectric loss of the composite ceramics with x = 0.3 and x = 0.4, possibly due to the presence of more pores. The dielectric loss is typically described by the following Debye formula [25,26]:

$$D = D_P + D_G = \frac{(\varepsilon_S - \varepsilon_\infty)\omega\tau}{\varepsilon_S + \varepsilon_\infty\omega^2\tau^2} + \frac{\gamma}{\omega\varepsilon_0}\frac{1}{\varepsilon_\infty + \frac{\varepsilon_S - \varepsilon_\infty}{1 + \omega^2\tau^2}} \tag{1}$$

where $D$ represents total dielectric loss, $D_P$ represents dielectric polarization loss, $D_G$ represents conductor conductivity loss, $\tau$ and $\gamma$ represent relaxation time and conductivity, respectively, $\varepsilon_S$ represents the static dielectric constant, $\varepsilon_\infty$ represents the optical frequency dielectric constant, $\varepsilon_0$ the represents vacuum dielectric constant, and $\omega$ represents frequency.

The total dielectric loss comprises two components: dielectric polarization loss and conductor conductivity loss. Dielectric polarization loss arises from the energy loss due to molecular movement under an alternating electric field, while conductor conductivity loss is caused by energy loss due to the resistance of the current within the material [27,28]. According to Formula (1), at very low frequencies where ω approaches 0, dielectric polarization loss (D_P) tends to 0, and the total dielectric loss is solely attributed to conductor conductivity loss. Furthermore, when the relaxation time ($\tau$) is much larger than the frequency (ω), $\omega\tau$ is significantly less than 1. Under such circumstances, the total dielectric loss can be simplified as follows [29,30]:

$$D \cong D_G \cong \frac{\gamma}{\omega\varepsilon_0\varepsilon_S} \tag{2}$$

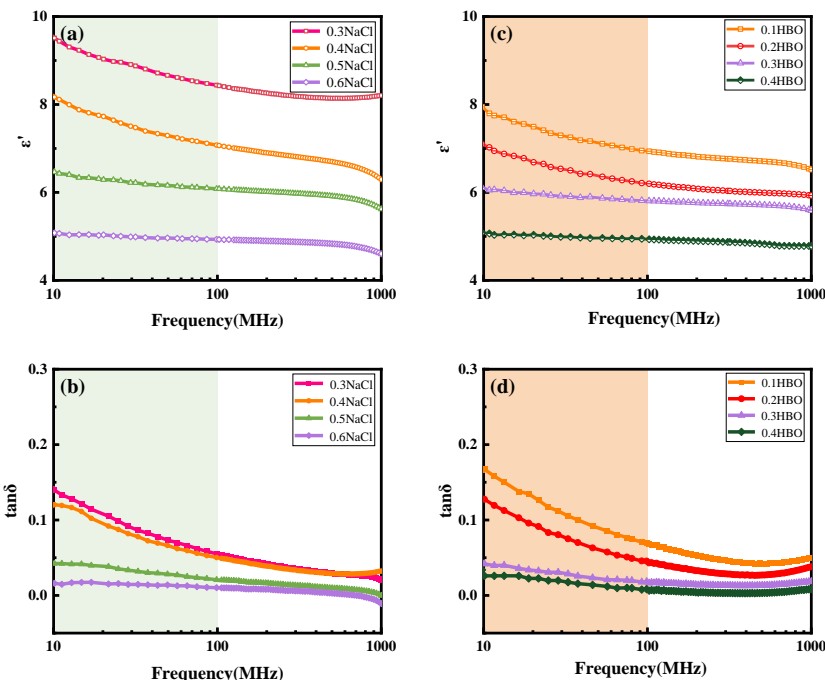

**Figure 3.** (**a**) Dielectric constant and (**b**) dielectric loss of xNaCl-(1−x)NZO; (**c**) dielectric constant and (**d**) dielectric loss of $xH_3BO_3$-$(0.8−x)Ni_{0.5}Zn_{0.5}Fe_2O_4$ composite ceramics.

According to Formula (2), the dielectric loss (*D*) is proportional to the conductivity ($\gamma$). In this experiment, the increase in NaCl content in composite ceramics will affect the conductivity of the material. Firstly, as the frequency rises, tanδ significantly decreases for all samples. According to the relaxation ion theory, electric charges locally accumulate under the action of an electric field due to the presence of relatively high-conductivity crystal grain phases in the insulating grain boundary matrix, resulting in interface polarization. It is well known that the polarization hysteresis in the alternating electric field determines the tanδ of inhomogeneous dielectric materials. The change in the loss curve shows that the attenuation of tanδ at higher frequencies is due to the resistive layer formed by the interfacial polarization and the hysteresis of the high-frequency polarization [31]. From Figure S1, it can be seen that with the increase in NaCl content, grain distribution becomes more uniform, and the outline of grain boundaries becomes clearer, indicating that the composition of high resistivity grain boundaries expanding, thereby reducing conductivity [32]. It is also worth noting that the changes between tanδ and relative density show a quite similar trend. When x varies from 0.3 to 0.6, the relative density increases from 87% to 96%. Optimization of relative density results in a reduction in porosity, improves the order of crystal structure and weakens the anharmonic vibration of the chemical bond, which is conducive to the reduction in dielectric loss; thus, it can be seen that the dielectric properties are jointly influenced by intrinsic and extrinsic factors.

Figure 3c presents the dielectric constant curves of HB-NZO-NaCl complex ceramics with varying HB contents. Figure 3d displays the tangent of the dielectric loss, which exhibits a similar trend to that of xNaCl/(1−x)NZO complex ceramics. However, the dielectric constant slightly decreases, which can be attributed to the relatively small dielectric constant value of $H_3BO_3$. At the same time, it can be observed that both dielectric constant and dielectric loss of HB-NZO-NaCl change more smoothly in the range of 10 MHz to 100 MHz than that of sample xNaCl/(1−x)NZO, indicating that HB-NZO-NaCl samples can maintain dielectric stability in a wide band. In addition, for the HB-NZO-NaCl sample, when x change from 0.3 to 0.4, the dielectric loss is always less than 0.05 in the measurement range, which shows that the loss of composite ceramics formed by adding $H_3BO_3$ can be significantly reduced compared with those of pure nickel–zinc ferrite, and that there is a possibility of application in discrete devices [33]. It is worth noting that for the

HB-NZO-NaCl composite ceramics, it can achieve densification at a lower temperature, and shows a more competitive dielectric characters in broadband performance compared with the first sample, so the magnetic properties of the HB-NZO-NaCl will be discussed next.

Figure 4 shows the frequency distribution of the permeability of xNaCl/(1−x)NZO multiphase ceramics at 200 °C/30 min/500 MPa. With the increase in NaCl content, the permeability presents a trend of gradual decrease. The change in magnetic loss also follows the same rules. Figure 5 shows the magnetic permeability and magnetic loss of HB-NZO-NaCl composite ceramics.

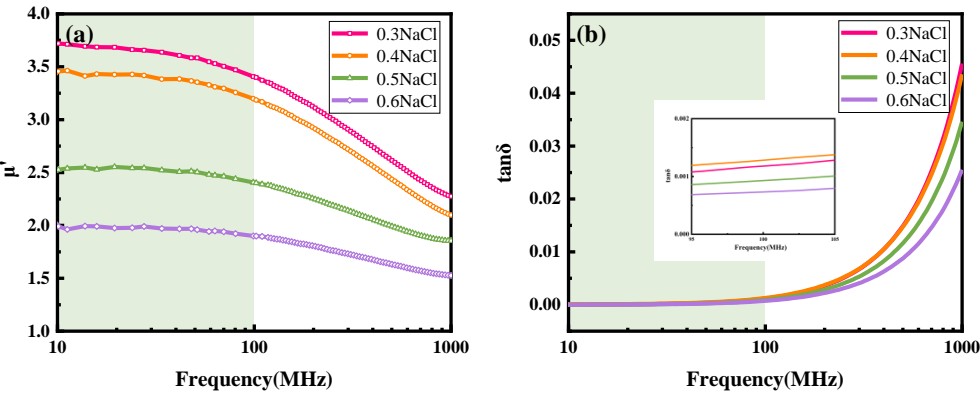

**Figure 4.** (**a**) Magnetic permeability and (**b**) magnetic loss of xNaCl-(1−x)NZO composite ceramics.

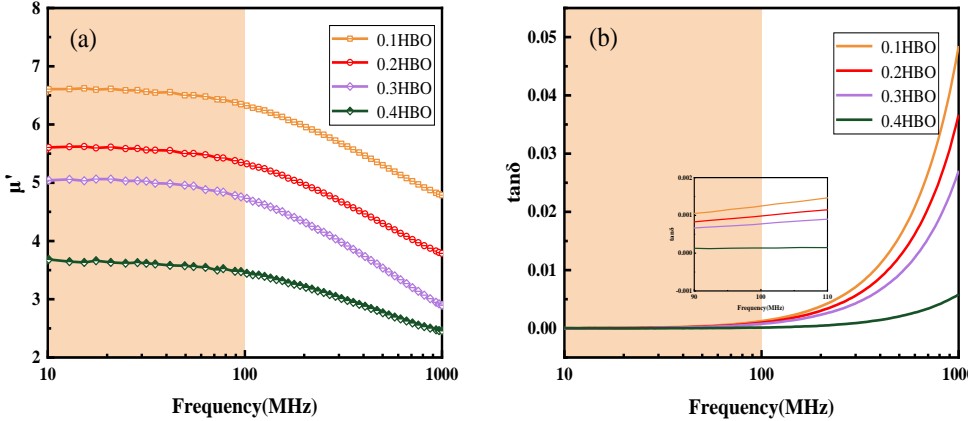

**Figure 5.** (**a**) Magnetic permeability and (**b**) magnetic loss of HB-NZO-NaCl composite ceramics.

Compared with the former sample, there are some obvious differences. The permeability of HB-NZO-NaCl composite ceramics is generally higher than that of xNaCl/(1−x)NZO, which is mainly affected by external factors [32]. It is certain that the grain size is larger for HB-NZO-NaCl samples, which is convenient to the domain movement [34]. The permeability is mainly caused by the displacement of the magnetic domain and deflection of the internal magnetic moments [35,36]. In comparison, the contribution of domain displacement to the permeability is more obvious. For both xNaCl/(1−x)NZO and HB-NZO-NaCl samples, the magnetic loss from 10 MHz to 100 MHz decreases with the increase in the non-magnetic component contents, while the magnetic loss decreases significantly between 500 MHz and 1000 MHz. This is due to the decay of the natural resonance phenomenon caused by the gradual reduction in the proportion of magnetic ions in the system. As for the HB-NZO-NaCl samples, when x = 0.4, the loss in the test range is always less than 0.006, which is obviously better than other samples.

Direct current (DC) resistivity is a significant characteristic of ferrite ceramics employed in electronic and magnetic devices. Figure 6 demonstrates that the DC resistivity increases as the content of the non-magnetic material HB rises. For instance, when x = 0.1,

the resistivity measures approximately $2 \times 10^8$ $\Omega \cdot$cm, while at an HB content of 0.4, the resistivity escalates to $3.4 \times 10^8$ $\Omega \cdot$cm. The dielectric structure of ferrite composite ceramics typically comprises two layers: ferrite grains exhibiting good conductivity and grain boundaries displaying poor conductivity [36]. With an augmentation in HB content, the proportion of grain boundaries with inadequate conductivity becomes more pronounced, consequently leading to higher resistivity in the magnetic–dielectric composite ceramics. In NZO ferrite, resistivity arises due to electron hopping between $Fe^{2+}$ and $Fe^{3+}$ ions [37]. However, in the context of cold sintering, lower temperatures have a minimal impact on the resistivity of the ferrite itself. A high resistivity is desirable for minimizing eddy current losses, which is beneficial for application as an insulation material in electrical equipment.

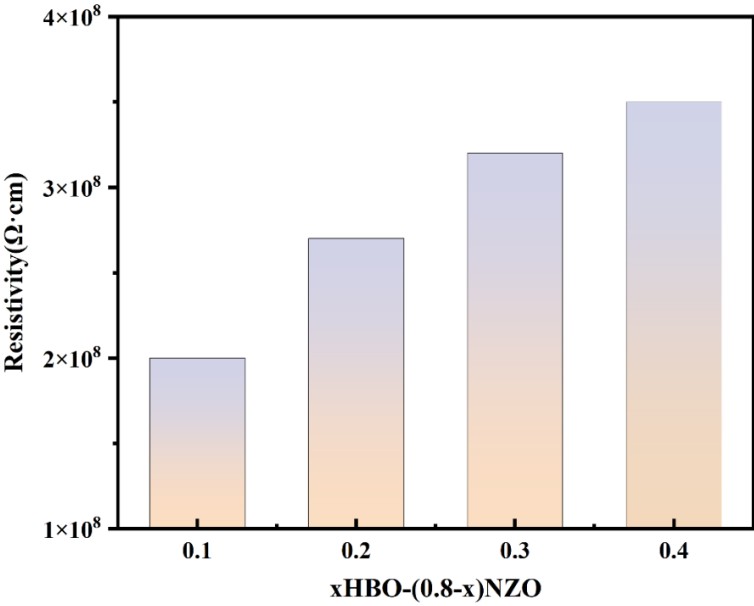

**Figure 6.** Electrical resistivity of HB-NZO-NaCl as a function of x values.

Figure 7 illustrates the frequency-dependent distribution of the magnetic permeability to dielectric constant ratio for the composite ceramic 0.3HB-0.5NZO-0.2NaCl. The plot reveals that within the frequency range of 10 MHz to 100 MHz, the magnetic permeability remains stable, ranging from 4.8 to 5.1, while the dielectric constant fluctuates between 5.9 and 6.1. Consequently, the ratio of magnetic permeability to dielectric constant varies from 0.83 to 0.85, exhibiting consistent behavior in this frequency range. This observation implies that the 0.3HB-0.5NZO-0.2NaCl composite ceramic has shown an isomagnetic dielectric effect in the range of 10 MHz to 100 MHz [29,33]. The material enables impedance matching across a wide frequency spectrum, thereby reducing signal reflection and loss while enhancing the transmission efficiency and performance of communication systems [38,39]. The physical image of 0.3HB-0.5NZO-0.2NaCl is displayed in Figure 8.

For isomagnetic dielectric materials, when used as discrete device substrates, they are expected to have a wide range of isomagnetic dielectric frequencies, reasonable values of dielectric constant and permeability, and minimal values of magnetic loss and dielectric loss [40]. Table 1 shows the performance comparison between this material and other isomagnetic dielectric materials reported previously. When the material is used for discrete devices, such as antenna substrates, the dielectric constant of the material should be guaranteed within a certain range. For low-frequency antennas, such as T-DMB antennas used in HF or VHF bands and built-in NFC antenna, if the dielectric constant of substrate is too small, it will not be possible to avoid excessive antenna volume. On the contrary, a higher dielectric constant will more easily excite surface waves, generate parasitic radiation modes, and reduce antenna radiation efficiency [41]. In addition, for some specific antenna types, when dielectric constant of the substrate is too high will increase the requirements

for machining accuracy, resulting in limited actual processing. The dielectric constant of $0.3H_3BO_3$-$0.2NaCl$-$0.5Ni_{0.5}Zn_{0.5}Fe_2O_4$ is higher than that of common substrates, and is controlled within 10. Therefore, compared with other isomagnetic dielectric materials, the dielectric constant of $0.3H_3BO_3$-$0.2NaCl$-$0.5Ni_{0.5}Zn_{0.5}Fe_2O_4$ is moderate. Simultaneously, this material has a relatively wide working frequency of isomagnetic dielectric, which can cover NFC, HF, and part of the VHF frequency ranges. More importantly, due to its significant magnetic loss advantage, $0.3H_3BO_3$-$0.2NaCl$-$0.5Ni_{0.5}Zn_{0.5}Fe_2O_4$ has the potential for application in mobile phone built-in NFC antennas, which is beneficial for dealing with complex metal environments in integrated applications, strengthening the magnetic field strength of antennas, and preventing the magnetic field of NFC antennas from being absorbed by metal substances, thereby improving communication induction distance.

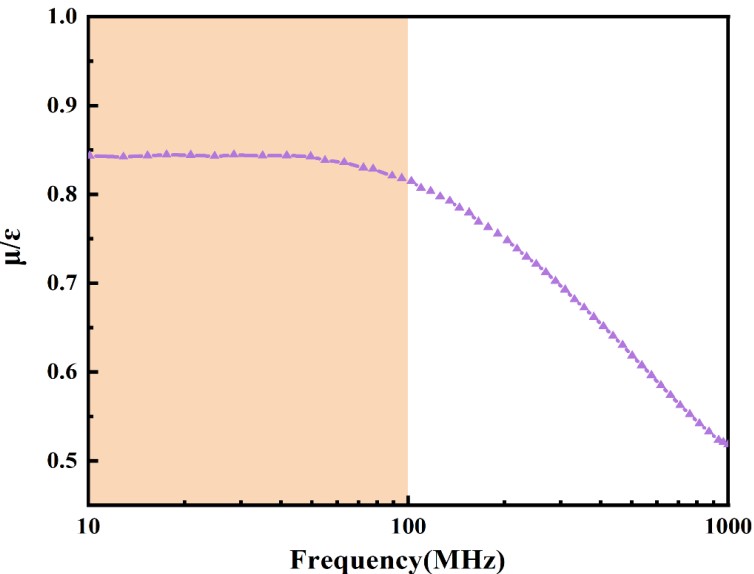

**Figure 7.** Distribution of the ratio of magnetic permeability and dielectric constant of composite ceramic $0.3H_3BO_3$-$0.2NaCl$-$0.5Ni_{0.5}Zn_{0.5}Fe_2O_4$ as a function of frequency.

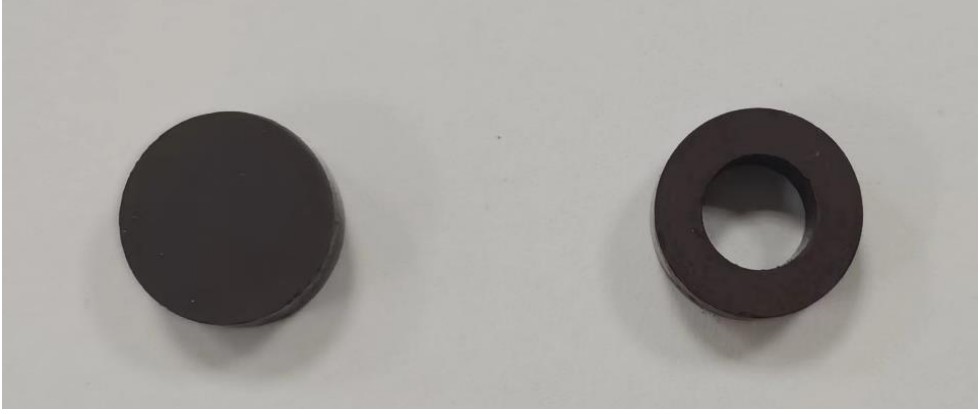

**Figure 8.** Vertical view of the $0.3H_3BO_3$-$0.2NaCl$-$0.5Ni_{0.5}Zn_{0.5}Fe_2O_4$ multiphase ceramic sample.

Consequently, it holds promise for addressing the challenges associated with free space matching and finding applications in wireless communication, filter design, and other related fields [48].

**Table 1.** Comparison of properties between different isomagnetic dielectric materials.

| | Dielectric Constant | Frequency Range (MHz) | $\tan\delta_\varepsilon$ (max) | $\tan\delta_\mu$ (max) | S.T (°C) | Ref. |
|---|---|---|---|---|---|---|
| $Mg_{0.96}Co_{0.04}Fe_{1.98}O_4 +$ 3 wt% $Bi_2O_3$ | ~10 | 3–30 | 0.04 | 0.04 | 1000 | [42] |
| $Ni_{0.855}Cu_{0.1}Zn_{0.025}Co_{0.02}Fe_{1.96}O_4$ | ~11.8 | 10–100 | 0.015 | 0.015 | 950 | [43] |
| $Li_{0.468}Co_{0.032}Fe_{2.484}O_4 +$ 3 wt% $Bi_2O_3$ | ~15 | 3–30 | 0.01 | 0.01 | 900 | [44] |
| $Ni_{0.5}Zn_{0.3}Co_{0.2}Fe_2O_4$ | ~5 | 100–500 | 0.02 | 0.04 | 950 | [45] |
| $(Ba_{0.5}Sr_{0.5})_3Co_2Fe_{24}O_{41} +$ 0.2 wt% $WO_3$ | ~12 | 10–200 | 0.002 | 0.05 | 1200 | [46] |
| $Ni_{0.368}Cu_{0.2}Zn_{0.432}Fe_{1.96}O_{3.94} +$ 12 wt% $TiO_2$ | ~12 | 1–40 | 0.0004 | 0.023 | 950 | [47] |
| $Ni_{0.35}Co_{0.02}Cu_{0.18}Zn_{0.45}Fe_{1.95}O_4$ + 15 wt% $BaTiO_3$ | ~28 | 3–30 | 0.002 | 0.02 | 1080 | [48] |
| $0.3H_3BO_3$-$0.2NaCl$-$0.5Ni_{0.5}Zn_{0.5}Fe_2O_4$ (this work) | ~6 | 10–100 | 0.036 | 0.001 | 200 | |

## 4. Conclusions

This study examines the feasibility of synthesizing $xNaCl$-$(1-x)Ni_{0.5}Zn_{0.5}Fe_2O_4$ and $xH_3BO_3$-$(0.8-x)Ni_{0.5}Zn_{0.5}Fe_2O_4$-$0.2NaCl$ composite ceramics using the cold sintering process. The investigation focuses on evaluating the impact of different cold sintering conditions on the sintering behavior as well as the magnetic and dielectric properties of the composite ceramics. Among the investigated compositions, the 0.2NaCl-0.3HB-0.5NZO composite exhibits promising characteristics, including a dielectric constant of 5.9 at 100 MHz, a dielectric loss of 0.02 at 100 MHz, a magnetic permeability of 5.1 at 100 MHz, and a magnetic loss of $5 \times 10^{-4}$ at 100 MHz. Notably, the composite material demonstrates similar dielectric and magnetic properties, with low losses, as well as stable performance within the frequency range of 10 MHz to 100 MHz. The simplicity of the design approach, coupled with the excellent magnetic and acceptable dielectric properties, highlights the potential of this composite material as a promising candidate for device fabrication.

**Supplementary Materials:** The following supporting information can be downloaded at: https://www.mdpi.com/article/10.3390/cryst13071140/s1. Figure S1. Sectional surface SEM diagram of multiphase ceramics xNaCl/(1−x)NZO (a) x = 0.3; (b) x = 0.4; (c) x = 0.5; (d) x = 0.6, Figure S2. (a) SEM diagram of 0.5NaCl/0.5NZO multiphase ceramics at 200 °C/30 min/500 MPa; (b) EDS mapping; (c) Na; (d) Cl; (e) Zn; (f) Ni; (g) Cu; (h) Fe. Figure S3. XRD pattern of different content of H3BO3. Figure S4. Sectional surface SEM figure of xH3BO3/(0.8−x)Ni0.5 Zn0.5 Fe2O4/0.2 NaCl (a) x = 0.1; (b) x = 0.2; (c) x = 0.3; (d) x = 0.4, Table S1. Dielectric loss of NaCl-NZO and HB-NZO-NaCl samples near some typical frequencies. Table S2. Magnetic loss of NaCl-NZO and HB-NZO-NaCl samples near some typical frequencies.

**Author Contributions:** Writing-original draft, J.H.; methodology, M.C.; validation B.L. and M.M.; investigation, H.B.B. and E.T.-N.; resources Z.M., Y.R. and F.S.; writing-review & editing, L.Y., D.W. and K.S. All authors have read and agreed to the published version of the manuscript.

**Funding:** This research was funded by [Natural Science Foundation of China] grant number [52161145401, 51672063]; [the Department of Science and Technology of Zhejiang province on the "sharp soldiers" and "leading geese" research and development research planning project] grant number [2023C01183] And The APC was funded by [Hangzhou Dianzi University].

**Data Availability Statement:** Data is contained within the article or Supplementary Material. The data presented in this study are available.

**Acknowledgments:** This work was supported by the Natural Science Foundation of China (Grant No.52161145401, 51672063), and the Department of Science and Technology of Zhejiang province on the "sharp soldiers" and "leading geese" research and development research planning project (No.2023C01183).

**Conflicts of Interest:** The authors declare that they have no known competing financial interest or personal relationships that could have appeared to influence the work reported in this paper.

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
