# Peer review of "Cold Sintering Isomagnetic Dielectric NaCl-Nickel Zinc Ferrite Composite Ceramics"

_crystals, doi:10.3390/cryst13071140_

Round 1
Reviewer 1 Report
The article "Cold sintering isomagnetic dielectric NaCl-Nickel zinc ferrite composite ceramics" discusses interesting aspects of the use of ceramic composites. Nevertheless, the description of the obtained results is too short and insufficient. It would be necessary to add photos of the tested materials as well as to deepen the analysis of the results obtained. In the current form, there is only a description of the data in the graphs. I recommend major revision.
Comments
1.Please complete the introduction with recent literature. Especially regarding the application of the described composites in electronics. Eg:
AC measurements, impedance spectroscopy analysis, and magnetic properties of Ni0.5Zn0.5Fe2O4/BaTiO3 multiferroic composites, 2022, https://doi.org/10.1016/j.mseb.2022.116025.
A state-of-the-art review on advanced ceramic materials: fabrication, characteristics, applications, and wettability, 2023, https://doi.org/10.1108/PRT-12-2022-0144
Comparative study of cold assisted and conventional sintering of (1-2x) K0.5Na0.5NbO3-xBaTiO3-xBiFeO3 multiferroic ceramics, 2023, https://doi.org/10.1016/j.mseb.2023.116632.
2.Experimental procedure; Please provide the trade names of the raw materials used, describe the calcining process
3.Fig. 1d is illegible, please extract it separately
4. Please describe in the text where equations (1) and (2) were used.
Please list the dielectric loss values in the table
5. Regarding Fig.6, the discussion of the results is too weak.
Minor editing of English language required
Reviewer 2 Report
Dear authors, thank you for this very nice paper that I've read with a great interest.
Here comes my comments to hopefully improve it even more:
In general, you should look over "punctuation", i.e. to addspaces after points and commas.
Abstract: In the abstract you say that these materials are for RF and 5G devices, however according to 5G frequency schemes, the sub-6GHz band works at about 3GHz and none of your measurements are made in that band. Please comment?
You never say what Er, µr, and loss values are aimed for. Since they are changing with frequency, it would be nice to have an indication of what you're aiming at and what are the values of competitor materials.
You claim your materials are "low loss", this is maybe true for your HB-NZO-NaCl magnetic losses, but dielectric losses are quite high compared to regular LTCC or many PCB materials, so in order t convince me you have to give comparative data.
For the xNaCl-(1-x)NZO you have quite a large change in Er and also loss tan close to the upper frequency limit, so again, what will happen even higher up in frequency?
In the text explaining Figure 1, I have a hard time to understand if you did pressing before sintering as stated on line 99, or applied pressure during sintering as stated on line 102. Also in the graphs of figure 1, when varying pressure, figs a) and c), is the temperature fixed here? to room temperature ? and for the graphs b) and d) is this after applying pressure or at the same time and what was that pressure? Following on line 125, you present temperature/time/pressure as a "whole" but are these parameters applied simultaneously or not? A clear presentation of the procedure would be helpful here.
Figure 2, please provide graphs a) and c) with same scales as well as c) and d) for easier comparison, this will help me to agree on what you claim in lines 190-192, beacaus with the graphs today, I'm not sure to see this.
If you could place Figures 3 and 4 closer, comparison would be easier. Again, during the µr discussion, I would appreciate to know what values you are aiming at and what can be done by other materials.
Figure 6, lines 248-251: here you say that µr/Er should be kept constant to be advantageous for efficient transmission and reception of EM waves. Does this then mean that at frequencies > 100 MHz, your material is not good any more?
Thank you very much and good luck in your future research!
Round 2
Reviewer 1 Report
The authors introduce the recommended corrections. The article may be accepted for publication.
Minor editing of English language required